# Calculation Method of DC Fault Overvoltage Peak Value for Multi-Send HVDC Systems with Wind Power

Fan Li [1], Dong Liu [1], Xiaonan Han [1], Boyu Qin [2,*], Zhongjian Liu [1], Haoyang Yu [1], De Zhang [3,4], Xiaofan Su [3,4] and Mingjie Wang [5]

1  State Grid Economic and Technological Research Institute Co., Ltd., Beijing 102209, China; lifan658@163.com (F.L.); liudong@chinasperi.sgcc.com.cn (D.L.); hanxiaonan@chinasperi.sgcc.com.cn (X.H.); liuzhongjian@chinasperi.sgcc.com.cn (Z.L.); yuhaoyang@chinasperi.sgcc.com.cn (H.Y.)
2  Department of Electrical Engineering, Xi'an Jiaotong University, Xi'an 710049, China
3  Hunan Key Laboratory of Energy Internet Supply-Demand and Operation, Changsha 410082, China; zhangd11@hn.sgcc.com.cn (D.Z.); suxf1234@126.com (X.S.)
4  State Grid Hunan Electric Power Company Limited Economic & Technical Research Institute, Changsha 410029, China
5  Power Dispatching Control Center of Guizhou Power Grid Co., Ltd., Guiyang 550002, China; wangmingjie@stu.xjtu.edu.cn
*  Correspondence: qinboyu@xjtu.edu.cn; Tel.: +86-180-9205-7836

**Abstract:** Commutation failure (CF) and DC blocking (DCB) faults are common occurrences in high-voltage direct current (HVDC) systems, and their impact on the power grid can be significant due to sudden power fluctuations. These issues pose particular challenges in multi-send HVDC systems due to the intricate interaction between AC and DC components. To tackle these challenges, this paper proposes a method for analyzing the peak overvoltage at the converter bus resulting from DC faults in multi-send HVDC systems. The proposed method comprehensively considers the influence of DC/DC coupling and wind turbine low-voltage ride through (LVRT) characteristics on overvoltage. It offers a straightforward approach to calculate the peak overvoltage following a DC fault without the need for complex modeling or dynamic simulation software. By leveraging the equivalent parameters of the AC system and operational parameters of the DC system, the method effectively quantifies the overvoltage. The primary objective of this study is to address multi-send HVDC systems and establish computational formulas that enable a quantitative assessment of transient overvoltage resulting from DC faults. The analysis explores several influencing factors, uncovering that fault-induced overvoltage is influenced by aspects such as system strength and wind turbine reactive power dynamics. In a single-send HVDC system, the level of overvoltage in the system is primarily affected by the short-circuit ratio. A higher short-circuit ratio results in a lower overvoltage level. On the other hand, in multi-send HVDC systems, the overvoltage level is determined by the equivalent impedance of the individual systems. In DC systems where turbines are present in the DC near zone, the overvoltage level at the converter bus is influenced by the power characteristics of the turbines during the LVRT. To validate the accuracy of the proposed method, a comprehensive verification process is conducted. Through this research, the paper aims to contribute to the understanding and management of transient overvoltage in multi-send HVDC systems. By considering relevant factors and employing an equivalent model, the proposed method offers a practical approach for assessing overvoltage and facilitating the design and operation of such systems.

**Keywords:** overvoltage; commutation failure; HVDC system; DC blocking

## 1. Introduction

HVDC transmission technology exhibits remarkable advantages, including its vast transmission capacity [1,2]. It proves particularly beneficial for long-distance power grid transmission [3]. However, one notable drawback of HVDC transmission is its significant

consumption of reactive power, often amounting to approximately half of the transmitted active power [4]. This issue is commonly addressed through the utilization of filters and power compensators [5]. A typical fault that may occur in HVDC transmission systems is commutation failure [6]. When such a failure arises, it causes substantial fluctuations in the system's voltage and reactive power. Consequently, it can result in DC locking faults and pose a threat to the stability of the power grid [7]. In particular, in multi-send HVDC systems, commutation failure-induced power fluctuations between multi-send HVDC transmissions may intensify the occurrence of overvoltage and other faults [8]. A study analyzing the transient overvoltage of a bipolar system is conducted in [9], and the behavior of equipment under different overvoltage conditions is examined in [10].

There are various forms of faults in DC transmission systems, with commutation failure and DC locking being prominent issues leading to overvoltage problems. Commutation failure results in a sharp drop in DC power, and if it occurs repeatedly, it can lead to DC locking. Such occurrences have a substantial impact on the AC grid [11]. Following a commutation failure fault, the control link within the DC system increases the trigger angle to delay the current rise, which causes a rapid increase in reactive power absorption by the system. Consequently, the voltage at the rectifier side experiences a significant decline [12]. Furthermore, due to the presence of a low-voltage current limiting link, the DC current diminishes rapidly, resulting in a substantial reduction in the reactive power absorbed by the rectifier system [13]. Consequently, the reactive power compensation capacitor in the rectifier system generates a considerable amount of residual reactive power, leading to overvoltage at the rectifier bus [14]. Moreover, in the event of a DC blocking fault, the active power consumed by the DC system rapidly drops to zero, and the resulting residual reactive power significantly elevates the AC voltage at the rectifier end [15,16]. The AC voltage gradually decreases once the reactive power compensation capacitor is disconnected [17,18].

According to [19], overvoltage resulting from commutation failure and DC blocking faults is primarily influenced by the system's short-circuit capacity and residual reactive power. Another ref. [20] describes how the level of overvoltage is closely associated with factors such as the short-circuit capacity and the timing of the safety control cutter action. Improper coordination can likely lead to a secondary increase in voltage. Moreover, ref. [21] categorizes the voltage dynamic process into two phases and asserts that converter fault overvoltage is affected by the system's short-circuit ratio. Furthermore, the high/low-voltage ride-through characteristics of wind turbines also impact the magnitude of converter fault overvoltage. Generally, when wind farms are closer to the converter bus, the transfer of reactive power from the wind farm to the faulty AC bus becomes easier. Addressing this, another ref. [22] explores the mechanism behind overvoltage caused by commutation failure and DC blocking, as well as the influence of wind turbine LVRT characteristics. It reveals that commutation failure induces a low-voltage state in turbines located in the vicinity, consequently exacerbating the overvoltage problem. Conversely, ref. [23] examines the mechanism of high-voltage off-grid wind turbines and the role of thermal turbines in supporting voltage.

Hence, these factors necessitate consideration when evaluating the grid-connected capacity of wind power [24]. Particularly, when integrating a large-scale wind power system into the delivery side, fault overvoltage characteristics emerge from the interaction between the transient properties of the DC system and the LVRT characteristics of the wind farm [25]. Therefore, a comprehensive examination of the LVRT characteristics of wind power clusters is essential for a better understanding of the fault overvoltage issue in the DC system. Such comprehensive analysis facilitates an enhanced comprehension of the fault overvoltage problem [26]. In the case of multi-send HVDC systems, the causes of commutation failure in the system become more intricate due to the presence of multiple rectifier stations in the region and electrical coupling among multiple DCs [27]. Additionally, the causes of failure exhibit greater diversity owing to the existence of various potential causal factors [28]. Currently, the literature concerning converter failure in multi-send HVDC systems does not

adequately reflect the impact of the multi-send structure on the converter failure-induced overvoltage problem [29–31]. Furthermore, the mechanism underlying the overvoltage problem induced by converter failure in a multi-send structure remains undisclosed, with the related literature and data being outdated [32,33], and some research findings not being updated in real-time. Consequently, it becomes imperative to propose an overvoltage assessment methodology for a multi-send HVDC system incorporating wind turbines, aiming to provide a clear direction for future research endeavors.

Regarding the rectifier side of the HVDC transmission system connected to wind power, the dynamics of wind power's reactive power also influence overvoltage. However, ref. [34] takes into account the active and reactive power dynamics of wind turbines and proposes an iterative calculation method for obtaining transient overvoltage. Nonetheless, this approach is time-consuming and fails to consider the impact of LVRT characteristics of wind turbines on overvoltage. Ref. [35] establishes a detailed analytical model for transient reactive voltage during multi-mode switching of wind turbines. Nevertheless, the modeling method is overly complex, impeding quick analysis of the overvoltage level. Both low- and high-penetration characteristics of the disturbed wind turbine generators (WTGs) can exacerbate overvoltage in the delivery system [36]. When examining the specific role of WTGs, ref. [34] considers the turbine's impact on grid operation as a reduction in short-circuit capacity. Furthermore, another study [36] investigates the scenario of wind power in the near zone on the rectifier side, considering near-zone LVRT following commutation failure in the HVDC system. In this analysis, the surplus reactive power of the turbine's reactive power compensation capacitor during the fault is introduced. The mechanism of system overvoltage generation at the sending end after converter failure is analyzed in a different study, which clarifies the influence of the turbine's LVRT reactive power characteristics on overvoltage. Additionally, through simulation, ref. [22] examines the mode of operation of low-penetration power characteristics and concludes that the wind farm produces additional reactive power delivery to the converter bus during LVRT, exacerbating the voltage issue. The analysis conducted thus far fails to account for the impact of the machine-end voltage on the wind farm's active power after the disturbance.

There is a limited number of studies that analyze the impact of multiple DC coupling on converter fault overvoltage. In [37], a calculation method is proposed for determining the converter bus overvoltage in a multi-send HVDC system based on the interaction factor of the sending voltages. However, the method yields a large error, as it fails to consider the reactive power flow during the fault. Therefore, it is necessary to conduct in-depth research on the effects of multiple DC coupling and turbine reactive power characteristics on converter fault overvoltage. Additionally, there is a need to develop quantitative assessment methods to evaluate overvoltage in turbine-connected multiple DC feeder systems after a converter fault. It is important to note that most existing studies primarily focus on analyzing the peak of the converter fault overvoltage and do not account for the voltage crossing integral safety constraint. Consequently, a more comprehensive evaluation of the system's voltage safety level following a converter fault has not been adequately addressed.

Therefore, this paper builds upon the existing HVDC model and takes into account the mutual coupling between multiple DC transmissions. It conducts calculations and derives an analytical expression for the transient overvoltage of the multi-send HVDC system, while also analyzing the impact of overvoltage in such a system. The analysis demonstrates that the transient overvoltage of a multi-terminal HVDC system is influenced by factors such as the strength of the AC system, the electrical distance between two systems, and the DC power. Ultimately, the accuracy of the obtained formula is validated through a simulation model.

To quantitatively assess transient overvoltage in a multi-send HVDC system integrated with a wind farm, it is crucial to conduct a comprehensive investigation into the dynamic response of DC-DC coupling and the LVRT capability of the wind farm.

DC-DC coupling is of paramount importance within the framework of a multi-send HVDC system, as it, along with AC-DC coupling, significantly influences the transient characteristics of the power system, as well as ensuring the security and stability of the sending HVDC system. Understanding the interplay between these coupling mechanisms is vital for accurately assessing and analyzing the system's behavior during fault events.

In the event that the voltage at the wind farm bus falls below a predetermined threshold value (typically around 0.9 p.u.) due to a fault in the nearby system, the wind farm engages in a low-voltage ride-through procedure. During this procedure, the current of the wind turbines is rapidly controlled to decrease within a specific range in order to prevent inverter over-current. To maintain the voltage within permissible limits, the wind farm adjusts its current proportionally to the magnitude of the voltage reduction. This control strategy ensures the safe and effective operation of the wind farm during low-voltage conditions.

Initially, the paper focuses on examining the analytical expression for transient overvoltage in a single-send HVDC system. Subsequently, it addresses the influence of DC-DC coupling and the LVRT capability of the wind farm, presenting a methodology for analyzing the disturbed transient voltage. The accuracy of this methodology is then verified through appropriate validation techniques. By exploring these key aspects, a comprehensive understanding of the transient overvoltage behavior in the integrated multi-send HVDC system with a wind farm can be achieved.

## 2. Analysis Method of Transient Overvoltage

### 2.1. Theoretical Analysis Method for DC Fault Overvoltage

The Three Norths region of China is internationally acclaimed for its abundant wind power resources, which have positioned it as a significant and valuable energy source. These vast wind power resources play a pivotal role in meeting the energy demands of the region and beyond. The region's strategic location allows for the transmission of this clean and renewable energy to the load center through a network of multiple HVDC systems. This extensive infrastructure ensures the efficient and reliable delivery of wind power, making the Three Norths region an indispensable contributor to the sustainable energy landscape. In the analysis of multi-send HVDC systems, the double-send HVDC system represents the most commonly employed unit. To streamline the analysis procedure, this paper employs a simplified model depicted in Figure 1, which approximates the large-scale wind power generation in the Three Norths region using a double-send HVDC system.

The system illustrated in Figure 1 represents a quasi-steady-state model, wherein all converter stations function in a rectified state, while the AC system is simplified to a fixed electric potential series reactance. The power on the rectifier side of the *i*th-sending HVDC system can be mathematically expressed by the following algebraic equation:

$$
\begin{cases}
P_{si} - P_{gsi} = 0 \\
P_{wi} - P_{gwi} = 0 \\
Q_{gsi} - Q_{xsi} - Q_{si} = 0 \\
Q_{gwi} - Q_{xwi} - Q_{wi} = 0 \\
P_{wi} + P_{si} - \sum_{j=1,j\neq i}^{n} P_{ij} - P_{dri} = 0 \\
Q_{si} + Q_{wi} + Q_{cpi} - Q_{dri} - \sum_{j=1,j\neq i}^{n} Q_{ij} = 0 \\
Q_{xwi} = \frac{P_{wi}^2 + Q_{wi}^2}{U_{pi}^2} X_{wi} \\
Q_{xsi} = \frac{P_{si}^2 + Q_{si}^2}{U_{pi}^2} X_i
\end{cases}
\tag{1}
$$

where $P_{gwi}$ and $Q_{gwi}$ are the active and reactive power, respectively, emitted by the turbine of AC system $i$; $P_{wi}$ and $Q_{wi}$ are the active and reactive power, respectively, delivered by the turbine side of AC system $i$; $P_{gsi}$ and $Q_{gsi}$ are the active and reactive power, respectively,

emitted by the equivalent synchronous machine of AC system $i$; $P_{si}$ and $Q_{si}$ are the active and reactive power, respectively, delivered by the system side of AC system $i$; $Q_{cpi}$ is the shunt compensated reactive power of converter station $i$; $P_{dri}$ and $Q_{dri}$ are the active and reactive power, respectively, consumed by DC system $i$; $U_{wi}$, $U_{si}$, and $U_{pi}$ are the fan-side, system-side, and converter station bus voltages, respectively, of AC system $i$; $X_{wi}$ and $X_i$ are the fan-side and system-side reactance, respectively, reflecting the electrical strength of system $i$; $X_{ij}$ is the impedance between system $i$ and system $j$, reflecting the strength of the connection between the two systems.

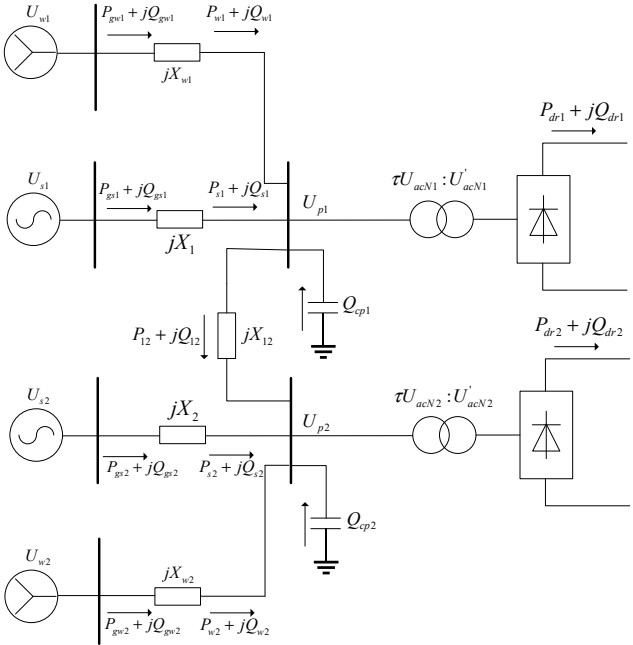

**Figure 1.** Simplified two-send DC system.

Figure 2 illustrates the voltage vector relationship between the system commutation bus, the system-side bus, and the fan-side bus after the fault, based on the line voltage loss formula. In the figure, $P'_{wi}$ and $Q'_{wi}$ are the active and reactive power, respectively, delivered from the fan side of AC system $i$ after the fault; $P'_{si}$ and $Q'_{si}$ are the active and reactive power, respectively, delivered from the system side of AC system $i$ after the fault; $P'_{ij}$ and $Q'_{ij}$ are the active and reactive power, respectively, delivered from system $i$ to system $j$ after the fault.

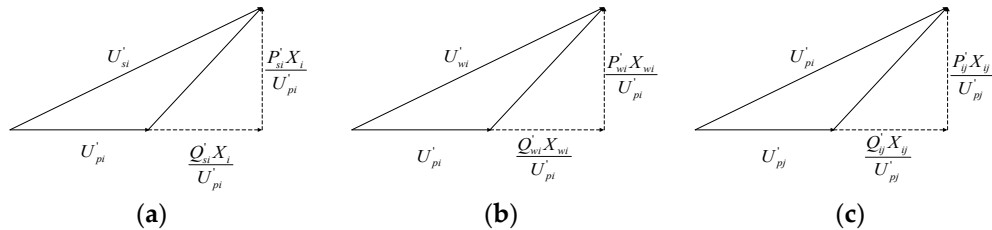

**Figure 2.** Voltage vector relationship diagram. (**a**) Converter station and system side; (**b**) Wind turbine and converter station; (**c**) Converter stations.

As seen in Figure 2a, the system-side voltage $U'_{si}$ can be obtained from the post-fault converter bus voltage $U'_{pi}$ and the post-fault system-side power delivered to the converter bus $P'_{si}$ and $Q'_{si}$ as follows:

$$Q'_{si} = -Q'_{gwi} + Q'_{xwi} - Q'_{cpi} + Q_{dri\_fault} + \sum_{j=1, j \neq i}^{n} Q'_{ij} \tag{2}$$

$$P'_{si} = -P'_{wi} + P_{dri\_fault} + \sum_{j=1, j \neq i}^{n} P'_{ij} \tag{3}$$

$$U'_{si} = \sqrt{\left(U'_{pi} + \frac{Q'_{si}X_i}{U'_{pi}}\right)^2 + \left(\frac{P'_{si}X_i}{U'_{pi}}\right)^2} \tag{4}$$

where $Q'_{gwi}$ is the turbine reactive power output after the fault; $Q'_{xwi}$ is the reactive power consumed on the turbine side after the fault; $Q'_{cpi}$ is the reactive power issued by reactive compensation after the fault; $P_{dri\_fault}$ and $Q_{dri\_fault}$ are the active and reactive power, respectively, consumed by the DC system after the fault.

From Figure 2b, the fan-side voltage $U'_{wi}$ can be obtained from the post-fault converter bus voltage $U'_{pi}$ and the post-fault fan-side delivered power $P'_{wi}$ and $Q'_{wi}$ as follows:

$$U'_{wi} = \sqrt{\left(U'_{pi} + \frac{Q'_{wi}X_{wi}}{U'_{pi}}\right)^2 + \left(\frac{P'_{wi}X_{wi}}{U'_{pi}}\right)^2} \tag{5}$$

From Figure 2c, the converter bus *i* voltage $U'_{pi}$ can be obtained from the post-fault converter bus *j* voltage $U'_{pj}$ and the post-fault fan-side delivered power $P'_{ij}$ and $Q'_{ij}$ as follows:

$$U'_{pi} = \sqrt{\left(U'_{pj} + \frac{Q'_{ij}X_{ij}}{U'_{pj}}\right)^2 + \left(\frac{P'_{ij}X_{ij}}{U'_{pj}}\right)^2} \tag{6}$$

AC filters and reactive power compensators primarily consist of capacitors, which exhibit an output reactive power that is directly proportional to the square of the voltage. Consequently, the expression for their output reactive power during a fault can be written as:

$$Q'_{cpi} = U'_{pi}{}^2 Q_{cpi} \tag{7}$$

The power generated by the wind turbine during LVRT is dependent on the bus voltage at the machine's end. The reactive power output of the unit is directly proportional to the magnitude of the voltage decrease, while the active power output is limited to a small value. The power emitted by the unit following a DC fault can be expressed as:

$$\begin{cases} Q'_{wi} = \lambda_{wi}\left(0.9 - U'_{wi}\right)Q_{gwi0} \\ P'_{gwi} = k_{wi}P_{gwi0} \end{cases} \tag{8}$$

where $\lambda_{wi}$ is the proportional factor of reactive power output during LVRT of turbines in the near zone of AC system *i*; $k_{wi}$ is the proportional factor of active power output during low-voltage ride-through of turbines in the near zone of AC system *i*; $P_{gwi0}$ and $Q_{gwi0}$ are the rated active and reactive power output, respectively, of turbines in AC system *i*.

To summarize, the bus overvoltage of each converter station following a DC fault can be determined by integrating Equations (2)–(8).

### 2.2. Overvoltage Peak Analysis Method Caused by DC Faults

In the event of a DC locking fault, it is assumed that the DC system undergoes a gradual cessation of operation, resulting in an eventual consumption of zero active and reactive power. It is understood that the system voltage reaches its maximum magnitude when the DC system completely ceases operation, signifying the occurrence of a DC locking fault overvoltage. During this scenario, the active and reactive power consumed by the DC system attains its peak value, as expressed below:

$$\left\{ \begin{array}{l} P_{dri\_DCB} = 0 \\ Q_{dri\_DCB} = 0 \end{array} \right. \tag{9}$$

The point at which the DC system experiencing commutation failure consumes the minimum amount of reactive power corresponds to the peak of the commutation failure overvoltage. At this moment, the active and reactive power consumed by the DC system can be mathematically represented as follows:

$$\left\{ \begin{array}{l} Q_{dri\_CF} = I_{di\min} \sqrt{U_{dri0}^2 - (I_{di\min} R_d)^2} \\ P_{dri\_CF} = U_{dri\min} I_{di\min} \end{array} \right. \tag{10}$$

where $U_{dri0}$ is the no-load DC voltage of the rectifier; $R_d$ is the DC resistance of the DC line; $I_{di\min}$ signifies the minimum DC current.

### 2.3. Analysis Method on Transient Overvoltage Cosidering LVRT of Wind Farm

During a fault in the DC system, nearby wind turbines can be affected. If the voltage at the turbine end falls below a predetermined threshold, the turbine enters a LVRT state. In this state, the active current is reduced, and the reactive current is controlled to prevent the inverter current from exceeding its limit and to maintain the turbine voltage above the set limit. The power output of the wind turbine during the LVRT process is influenced by the voltage at the end bus. The turbine's reactive power output is positively correlated with the magnitude of the voltage reduction, while the active power output is moderately controlled. Equations (2)–(8) can be used to express the turbine's reactive power output following a fault.

The LVRT characteristics of turbines are analyzed separately for DC blocking and commutation failure faults. In the case of DC blocking faults, the voltage at the commutation bus rapidly increases, and the turbine does not enter the LVRT state since the voltage at each bus remains above the LVRT threshold. Therefore, the excess reactive power sources during the fault only include the additional reactive power surplus caused by the blocking fault. Thus, there is no need to consider the impact of LVRT characteristics on overvoltage.

Regarding commutation failure, the reactive power absorbed by the converter station significantly increases, leading to a sharp decline in the converter bus voltage. Simultaneously, as the trigger angle increases, the DC current gradually decreases until it reaches zero. Consequently, the reactive power consumed by the converter station also drops to zero, resulting in overvoltage in the converter station and its surrounding region. During the fault, the voltage at the converter bus decreases to 0.6 p.u., triggering a widespread LVRT condition in wind power.

The LVRT characteristics of wind turbines result in an increase in the reactive power from the turbine and a decrease in the reactive power from the system side. This reactive voltage behavior further contributes to an elevation in the converter bus voltage. It is evident that the significant amount of turbine LVRT induced by the DC commutation failure becomes an additional source of reactive power, exacerbating the voltage rise, along with the remaining reactive power caused by the commutation failure.

Currently, national standards lack specific guidelines concerning the low-penetration characteristics of wind turbines. As a result, manufacturers produce turbine models with highly diverse low-penetration characteristics.

The GE doubly fed/directly driven model represents the fundamental model with standard power control for LVRT power characteristics. Conversely, the T1 doubly fed model serves as an advanced model with autonomous LVRT control connections, allowing flexible adjustment of various LVRT active and reactive power characteristics to align with the specific circumstances. The GE double-fed model corresponds to turbine type 10 in the PSASP (version 7.51.02) simulation software, the GE direct-drive model to turbine type 11, and the T1 double-fed model to turbine type 12. Table 1 provides an overview of the LVRT characteristics for different turbine models.

**Table 1.** Low-voltage ride-through characteristics of different wind turbine models.

| Wind Turbine Models | Low-Voltage Ride-through Active Characteristics | Low-Voltage Ride-through Reactive Power Characteristics |
|---|---|---|
| GE double-fed | $I_{pi} = 0.1$, slope recovery | $K_{vi} = 0$, $K_{qi} = 0.8$ |
| GE direct-drive | $I_{pi} = 2.44U_{wi} - 0.976$, rapid recovery | $K_{vi} = 120$, $K_{qi} = 0.1$ |
| T1 double-fed | $I_{pi} = 0.1$, slope recovery | $I_{qi} = 0.7(0.9 - U_{wi}) + 0.47$ |

In the table, $I_{pi}$ and $I_{qi}$ are the active and reactive currents of the turbine, respectively; $K_{vi}$ and $K_{qi}$ are the internal loop control voltage and reactive gain of the turbine, respectively.

During a fault, both the reduction in active power output and the increase in reactive power output of a wind turbine contribute to a voltage rise. Referring to Table 1, the order of active power output during LVRT is as follows: GE direct-drive > GE double-fed > T1 double-fed. As for reactive power output during LVRT, the order is T1 double-fed > GE direct-drive/GE double-fed. In practice, the GE double-fed, GE direct-drive, and T1 double-fed models are utilized in the respective feeder systems. Following a DC commutation failure, Figure 3 illustrates the active power output of the turbine, Figure 4 shows the reactive power output, and Figure 5 depicts the bus voltage. Figures 3–5 show the dynamic characteristics of active, reactive, and turbine port voltages during LVRT for three different types of turbines (GE double-fed, GE direct-drive, and T1 double-fed). The findings reveal that the T1 double-fed turbine exhibits the highest continuous reactive power output and the most pronounced fault overvoltage, while the GE double-fed and GE direct-drive turbines demonstrate lower reactive power output and smaller fault overvoltage compared to the T1 double-fed turbine. Furthermore, the reactive power output of the GE direct-drive turbine is lower than that of the GE double-fed turbine, and the fault overvoltage diminishes at a faster rate than that of the GE double-fed turbine.

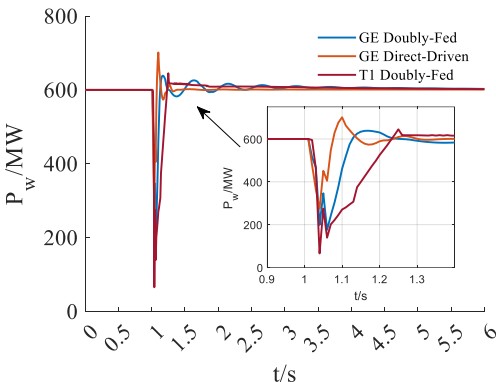

**Figure 3.** Active power output curves for all types of wind turbines.

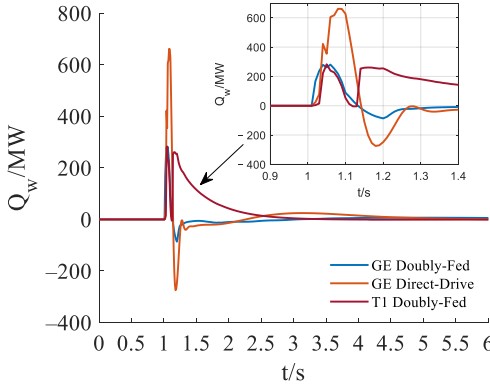

**Figure 4.** Reactive power output curves for all types of wind turbines.

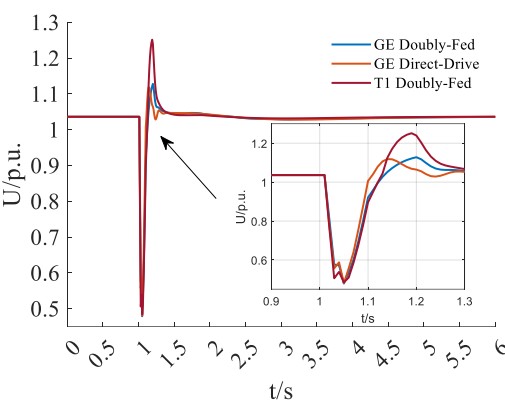

**Figure 5.** Wind farm-side voltage curve.

## 3. Case Study

### 3.1. Case Study of Transient Overvoltage in Single-Send HVDC System

A single-send HVDC system, comprising a sole HVDC transmission, can be referred to as an AC system. This system can be represented by an ideal voltage source connected in series with an equivalent impedance, resembling a Thevenin circuit. The analysis method is verified by constructing a simulation model in PSCAD, as depicted in Figure 6, where $P_{acr}$ denotes the active power transmitted by the AC system; $Q_{acr}$ denotes the reactive power transmitted by the AC system. The model is based on the CIGRE HVDC standard test model. The simulation parameters are set as follows: rated rectifier DC voltage $U_{drN} = 500$ kV; $R_d = 5$ Ω; rater converter bus voltage $U_{LN} = 345$ kV; the rated DC current is 4 kA; the rated DC power is 1000 MW.

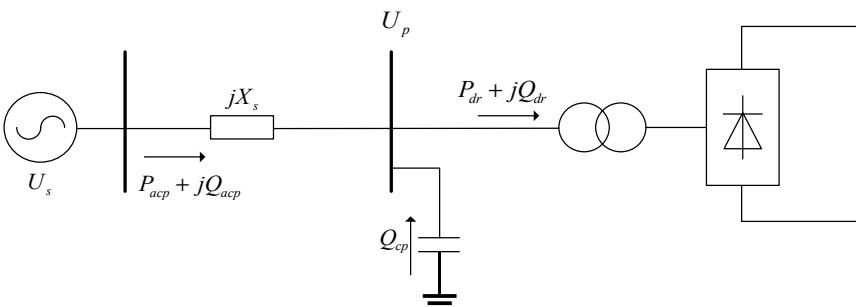

**Figure 6.** Single-send HVDC system.

The AC system is represented by an equivalent model, and its short-circuit capacity $S_{cr}$ can be adjusted by modifying the relevant parameters. The short-circuit capacity is equal to the square of the rated voltage divided by the equivalent impedance of the system. The short-circuit ratio is equal to the short-circuit capacity divided by the DC power rating. In single DC systems, the short-circuit ratio provides a good picture of the strength of the AC grid and has a clear physical concept, e.g., a system with a short-circuit ratio less than 2 is a very weak grid; a system with a short-circuit ratio greater than 3 is a strong grid; a system with a short-circuit ratio greater than 2 and less than 3 is a weak grid. This paper focuses on evaluating the strength of a power system, which assesses its capability to restore the voltage to its steady-state value following a disturbance.

To assess the impact of commutation failure on transient voltage, a 3-phase short-circuit fault is introduced at 3 s, lasting for 100 ms. Similarly, to investigate the influence of DC blocking on transient voltage, a DC blocking event occurs at 3 s. By varying the short-circuit capacity values, we can analyze the effect of this parameter on the overvoltage resulting from commutation failure and DC blocking. The simulation results for the overvoltage caused by commutation failure and DC blocking under different short-circuit capacity

values are presented in Tables 2 and 3, respectively. The expression for the error is the absolute value of simulation results minus calculation results divided by simulation results.

**Table 2.** Overvoltage peak value of commutation failure under different system short-circuit capacity.

|  | Case1 | Case 2 | Case 3 | Case 4 |
|---|---|---|---|---|
| $S_{Cr}$/p.u. | 2.86 | 3.33 | 4.00 | 5.00 |
| $X_s$/p.u. | 0.35 | 0.3 | 0.25 | 0.2 |
| $Q_{cp}$/p.u. | 0.453 | 0.397 | 0.290 | 0.199 |
| $Q_{acp}$/p.u. | 0.133 | 0.202 | 0.305 | 0.408 |
| $I_{dmin}$/p.u. | 0.2 | 0.2 | 0.2 | 0.2 |
| $U_{dr0}$/p.u. | 1.549 | 1.549 | 1.549 | 1.549 |
| Simulation results/p.u. | 1.147 | 1.124 | 1.085 | 1.062 |
| Calculation results/p.u. | 1.146 | 1.116 | 1.084 | 1.065 |
| error/% | 0.09 | 0.71 | 0.99 | 0.28 |

**Table 3.** Overvoltage peak value of DC blocking under different system short-circuit capacity.

|  | Case 5 | Case 6 | Case 7 | Case 8 |
|---|---|---|---|---|
| $S_{Cr}$/p.u. | 2.86 | 3.33 | 4.00 | 5.00 |
| $X_s$/p.u. | 0.35 | 0.3 | 0.25 | 0.2 |
| $Q_{cp}$/p.u. | 0.453 | 0.397 | 0.290 | 0.199 |
| $Q_{acr}$/p.u. | 0.133 | 0.202 | 0.305 | 0.408 |
| Simulation results/p.u. | 1.282 | 1.239 | 1.187 | 1.148 |
| Calculation results/p.u. | 1.324 | 1.245 | 1.177 | 1.133 |
| error/% | 3.28 | 0.48 | 0.84 | 1.31 |

The presented results shown in Figure 7 display the overvoltage occurrences resulting from commutation failure and DC blocking failure at various short-circuit capacities. It is observed that an increase in the short-circuit capacity corresponds to a lower level of system overvoltage. The simulation results are compared with the analytical results obtained from the mathematical model, revealing a negligible difference with an error of no more than 4%. These simulation outcomes serve as compelling evidence supporting the accuracy and effectiveness of the proposed calculation method for calculating the overvoltage peak value in this study.

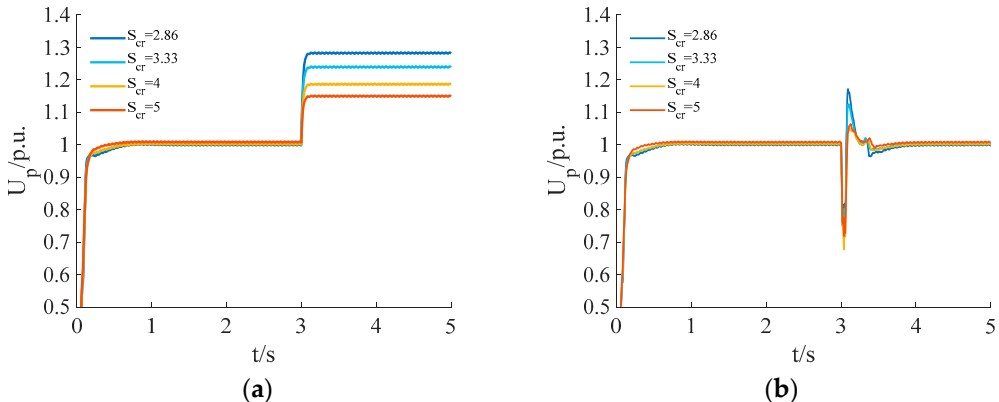

**Figure 7.** Verification results of single-send system. (**a**) DC blocking; (**b**) Commutation failure.

### 3.2. Analysis of the Effect of the Low-Voltage Ride-through Characteristics of the Wind Turbine on the Overvoltage Peak

To examine the impact of the wind turbine's LVRT characteristics on peak overvoltage following a failure, we utilize the PSD-BPA (version 2.7b) software to construct the feeder system model, as depicted in Figure 8. In this study, the T1 doubly fed turbine model as a representative example is employed.

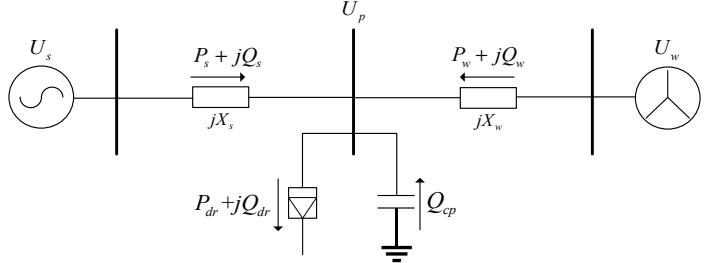

**Figure 8.** Two-machine equivalent system with wind farm.

In the event of a DC commutation failure, the turbines located in the proximity enter the LVRT mode, resulting in an increase in reactive power output. Conversely, during a DC blocking fault, the voltage at the converter station and its nearby area experiences a rapid rise, and the turbines do not undergo LVRT. A comparison of peak fault overvoltage for various numbers of turbines is presented in Table 4. Additionally, Figure 9 illustrates the post-fault voltage dynamic curve.

**Table 4.** Peak fault overvoltage in the near zone of the converter bus with different number of fans.

| Number of Fans/Unit | $X_s$/p.u. | Peak Value of CF/p.u. | | Error/% | Peak Value of DCB/p.u. | | Error/% |
|---|---|---|---|---|---|---|---|
| | | Simulation Results | Calculation Results | | Simulation Results | Calculation Results | |
| 300 | 0.12 | 1.150 | 1.147 | 0.26 | 1.098 | 1.091 | 0.64 |
| 600 | 0.12 | 1.201 | 1.193 | 0.67 | 1.100 | 1.091 | 0.82 |
| 900 | 0.12 | 1.249 | 1.257 | 0.64 | 1.098 | 1.091 | 0.64 |
| 1200 | 0.12 | 1.308 | 1.311 | 0.23 | 1.098 | 1.091 | 0.64 |

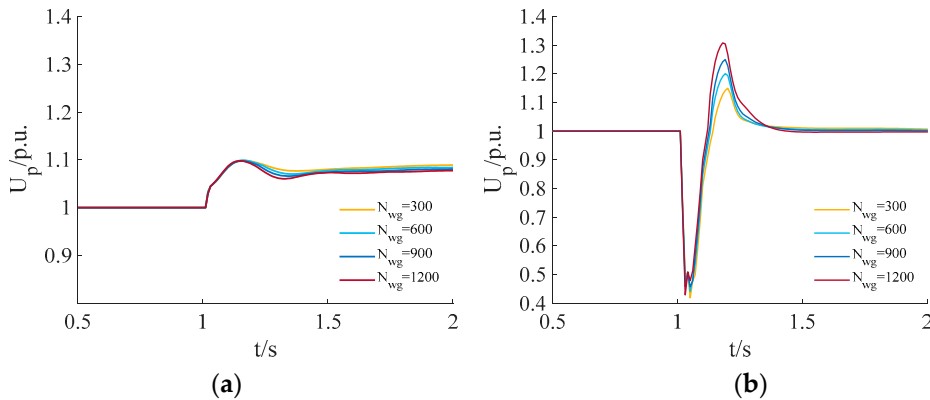

**Figure 9.** Verification results of system with wind farm. (**a**) DC blocking; (**b**) Commutation failure.

### 3.3. Case Study of Transient Overvoltage in Two-Sending HVDC System

PSCAD was utilized to construct a model of a two-send HVDC system, excluding turbines, aiming to analyze the impact of each equivalent impedance on the voltage of the converter bus (shown in Figure 10). The DC system model represents a single-pole HVDC system with a DC-side voltage of 500 kV and a capacity of 1000 MW, along with a rectifier-side AC system rated at 345 kV. The occurrence of a converter fault is set within system 1. The simulation results, as well as the model calculation results, illustrating the overvoltage induced by the converter fault under different system equivalent impedances are presented in Tables 5–10. Moreover, the voltage dynamics following a system fault are depicted in Figures 11–16.

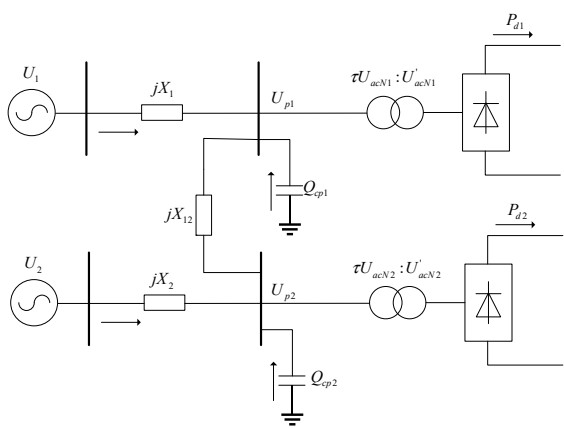

**Figure 10.** Two-send HVDC equivalent system.

**Table 5.** Different equivalent impedances $X_1$ DC blocking fault overvoltage peak.

| $X_1$/p.u. | $X_2$/p.u. | $X_{12}$/p.u. | Peak Value of CF/p.u. | | Error/% | Peak Value of DCB/p.u. | | Error/% |
|---|---|---|---|---|---|---|---|---|
| | | | Simulation Results | Calculation Results | | Simulation Results | Calculation Results | |
| 0.3 | 0.4 | 0.4 | 1.2034 | 1.2078 | 0.37 | 1.1010 | 1.1607 | 5.42 |
| 0.4 | 0.4 | 0.4 | 1.2539 | 1.2541 | 0.02 | 1.1406 | 1.1889 | 4.23 |
| 0.5 | 0.4 | 0.4 | 1.3012 | 1.2976 | 0.28 | 1.1745 | 1.2154 | 3.48 |
| 0.6 | 0.4 | 0.4 | 1.3302 | 1.3390 | 0.66 | 1.1894 | 1.2405 | 4.30 |

**Table 6.** Different equivalent impedances $X_1$ commutation failure overvoltage peak.

| $X_1$/p.u. | $X_2$/p.u. | $X_{12}$/p.u. | Peak Value of CF/p.u. | | Error/% | Peak Value of DCB/p.u. | | Error/% |
|---|---|---|---|---|---|---|---|---|
| | | | Simulation Results | Calculation Results | | Simulation Results | Calculation Results | |
| 0.3 | 0.4 | 0.4 | 1.0985 | 1.0907 | 0.71 | 1.0359 | 1.0571 | 2.05 |
| 0.4 | 0.4 | 0.4 | 1.1256 | 1.1658 | 0.36 | 1.0496 | 1.1039 | 5.17 |
| 0.5 | 0.4 | 0.4 | 1.1862 | 1.1925 | 0.53 | 1.1145 | 1.1204 | 0.53 |
| 0.6 | 0.4 | 0.4 | 1.2294 | 1.2183 | 0.90 | 1.1197 | 1.1363 | 1.48 |

**Table 7.** Different equivalent impedances $X_2$ DC blocking fault overvoltage peak.

| $X_1$/p.u. | $X_2$/p.u. | $X_{12}$/p.u. | Peak Value of CF/p.u. | | Error/% | Peak Value of DCB/p.u. | | Error/% |
|---|---|---|---|---|---|---|---|---|
| | | | Simulation Results | Calculation Results | | Simulation Results | Calculation Results | |
| 0.4 | 0.3 | 0.4 | 1.2398 | 1.2371 | 0.22 | 1.1146 | 1.1567 | 3.78 |
| 0.4 | 0.4 | 0.4 | 1.2539 | 1.2541 | 0.02 | 1.1406 | 1.1889 | 4.23 |
| 0.4 | 0.5 | 0.4 | 1.2739 | 1.2704 | 0.27 | 1.1693 | 1.2195 | 4.29 |
| 0.4 | 0.6 | 0.4 | 1.2818 | 1.2863 | 0.35 | 1.1884 | 1.2487 | 5.07 |

**Table 8.** Different equivalent impedances $X_2$ commutation failure overvoltage peak.

| $X_1$/p.u. | $X_2$/p.u. | $X_{12}$/p.u. | Peak Value of CF/p.u. | | Error/% | Peak Value of DCB/p.u. | | Error/% |
|---|---|---|---|---|---|---|---|---|
| | | | Simulation Results | Calculation Results | | Simulation Results | Calculation Results | |
| 0.4 | 0.3 | 0.4 | 1.1119 | 1.1522 | 3.62 | 1.0450 | 1.0791 | 3.26 |
| 0.4 | 0.4 | 0.4 | 1.1256 | 1.1658 | 3.57 | 1.0496 | 1.1039 | 5.17 |
| 0.4 | 0.5 | 0.4 | 1.1447 | 1.1790 | 3.00 | 1.0681 | 1.1279 | 5.60 |
| 0.4 | 0.6 | 0.4 | 1.1367 | 1.1921 | 4.87 | 1.0734 | 1.1512 | 7.24 |

**Table 9.** Different equivalent impedances $X_{12}$ DC blocking fault overvoltage peak.

| $X_1$/p.u. | $X_2$/p.u. | $X_{12}$/p.u. | Peak Value of CF/p.u. | | Error/% | Peak Value of DCB/p.u. | | Error/% |
|---|---|---|---|---|---|---|---|---|
| | | | Simulation Results | Calculation Results | | Simulation Results | Calculation Results | |
| 0.4 | 0.4 | 0.3 | 1.2533 | 1.2479 | 0.43 | 1.1641 | 1.1958 | 2.72 |
| 0.4 | 0.4 | 0.4 | 1.2539 | 1.2541 | 0.02 | 1.1406 | 1.1889 | 4.23 |
| 0.4 | 0.4 | 0.5 | 1.2712 | 1.2594 | 0.93 | 1.1390 | 1.1828 | 3.85 |
| 0.4 | 0.4 | 0.6 | 1.2782 | 1.2641 | 1.10 | 1.1302 | 1.1773 | 4.17 |

**Table 10.** Different equivalent impedances $X_{12}$ commutation failure overvoltage peak.

| $X_1$/p.u. | $X_2$/p.u. | $X_{12}$/p.u. | Peak Value of CF/p.u. | | Error/% | Peak Value of DCB/p.u. | | Error/% |
|---|---|---|---|---|---|---|---|---|
| | | | Simulation Results | Calculation Results | | Simulation Results | Calculation Results | |
| 0.4 | 0.4 | 0.3 | 1.1174 | 1.1597 | 3.79 | 1.0582 | 1.1103 | 4.92 |
| 0.4 | 0.4 | 0.4 | 1.1256 | 1.1658 | 3.57 | 1.0496 | 1.1039 | 5.17 |
| 0.4 | 0.4 | 0.5 | 1.1367 | 1.1711 | 3.03 | 1.0483 | 1.0981 | 4.75 |
| 0.4 | 0.4 | 0.6 | 1.1455 | 1.1758 | 2.65 | 1.0465 | 1.0929 | 4.43 |

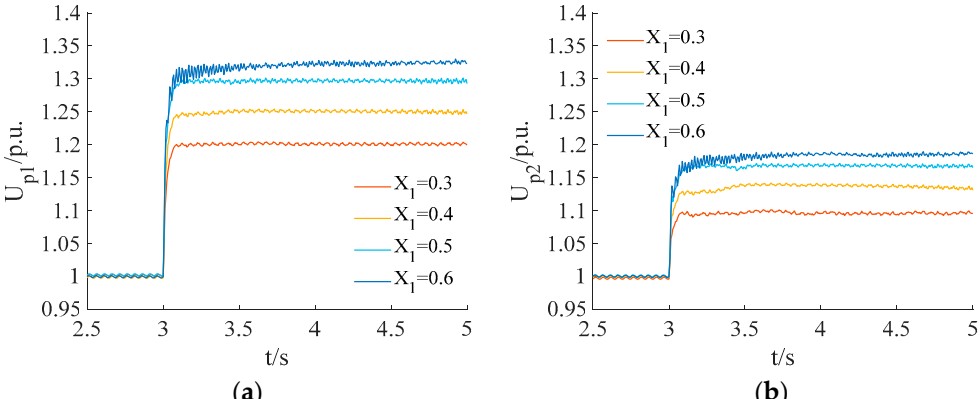

**Figure 11.** Bus voltage dynamics with different system equivalent impedance $X_1$ of DCB. (**a**) BUS 1; (**b**) BUS 2.

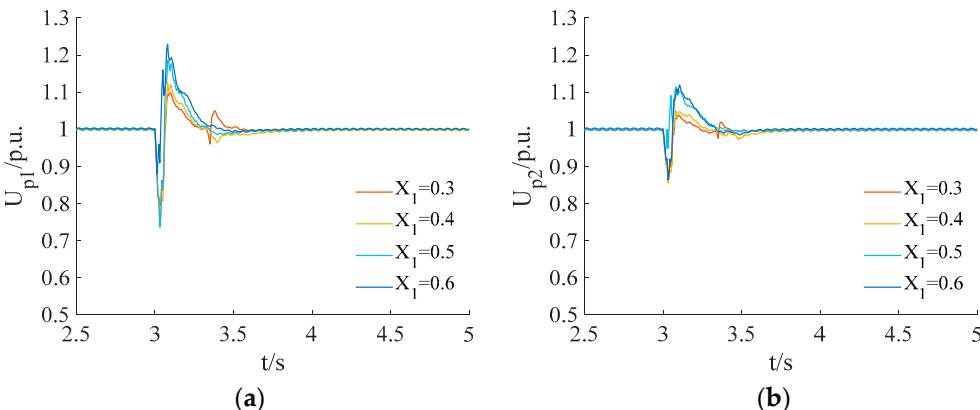

**Figure 12.** Bus voltage dynamics with different system equivalent impedance $X_1$ of CF. (**a**) BUS 1; (**b**) BUS 2.

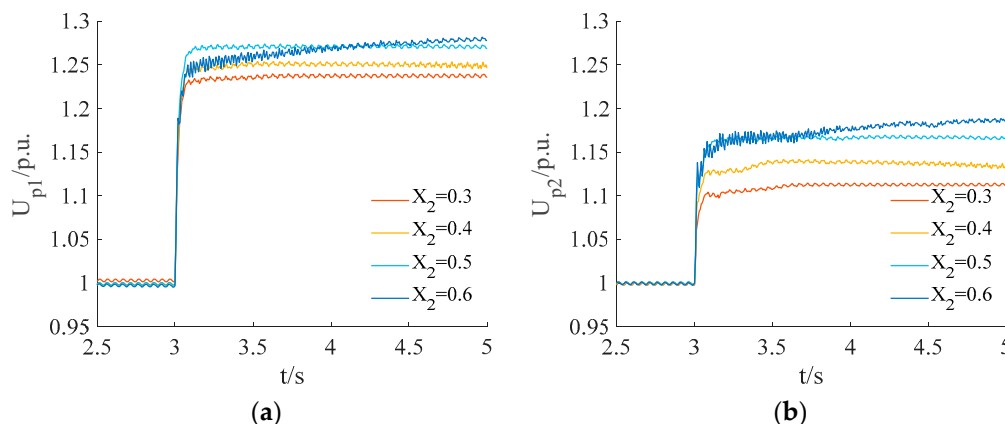

**Figure 13.** Bus voltage dynamics with different system equivalent impedance $X_2$ of DCB. (**a**) BUS 1; (**b**) BUS 2.

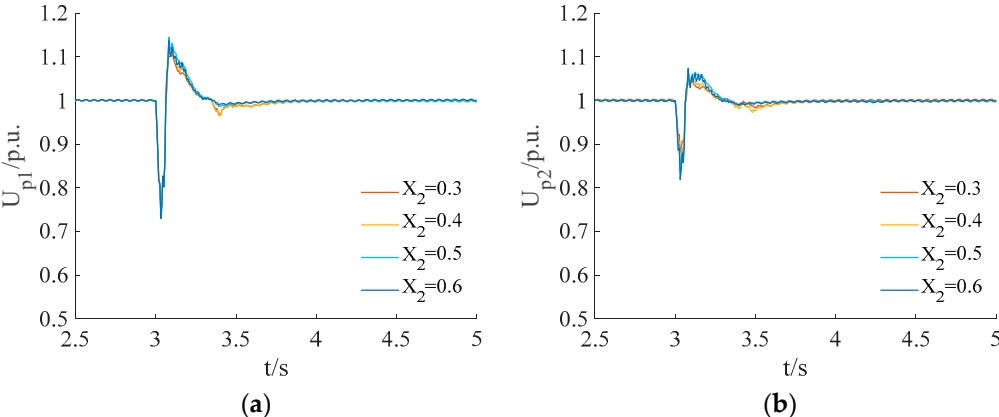

**Figure 14.** Bus voltage dynamics with different system equivalent impedance $X_2$ of CF. (**a**) BUS 1; (**b**) BUS 2.

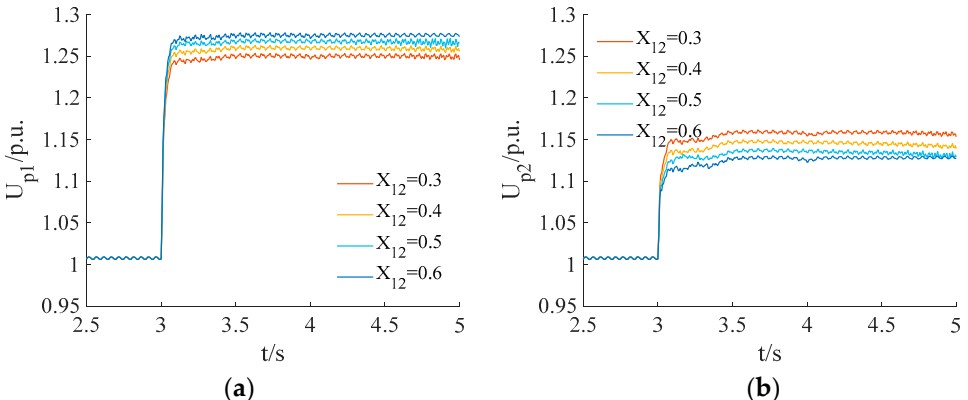

**Figure 15.** Bus voltage dynamics with different system equivalent impedance $X_{12}$ of DCB. (**a**) BUS 1; (**b**) BUS 2.

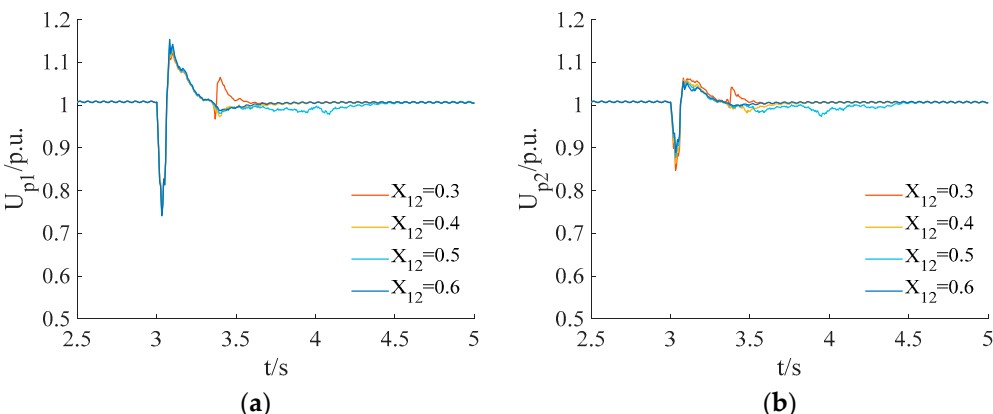

**Figure 16.** Bus voltage dynamics with different system equivalent impedance $X_{12}$ of CF. (**a**) BUS 1; (**b**) BUS 2.

The analysis reveals that in the event of a fault occurring in DC system 1, both DC buses experience an increase in overvoltage levels corresponding to the rise in system equivalent impedances $X_1$ and $X_2$. However, it is noteworthy that the equivalent impedance $X_1$ exerts a considerably greater influence on the system's peak overvoltage. Moreover, elevating the inter-system link impedance $X_{12}$ leads to an escalation in the overvoltage level of system 1's commutation bus, whereas system 2's overvoltage level diminishes. This phenomenon can be attributed to the reduction in reactive power transfer due to the weakening of the inter-system link's strength. Comparing the simulation results to the analysis results of the mathematical model reveals a minimal disparity between the two. The simulation results validate the precision and efficacy of the proposed overvoltage calculation method in this paper.

Fundamentally, when the system equivalent impedance experiences an elevation, it triggers a significant reduction in the system's overall strength. As a result, this diminished strength further exacerbates the intensification of overvoltage within the faulty system. Understanding the dynamics of overvoltage necessitates recognizing the pivotal relationship between system equivalent impedance and strength. With the increase in impedance, additional constraints are imposed upon the system's capacity to endure and regulate electrical disturbances, ultimately compromising its resilience. The adverse consequences of elevated equivalent impedance highlight the need for meticulous analysis and management of this parameter to mitigate the detrimental effects of overvoltage on the power system.

### 3.4. Case Study of Transient Overvoltage Considering LVRT of Wind Farm

In this section, the accuracy and applicability of the proposed method are verified using the PSD-BPA (version 2.7b) simulation software. The Northwest Power Grid project serves as the simulation object for this verification process. The HVDC transmission capacity of the system is rated at 3000 MW, and a GE doubly-fed turbine model is utilized; a geographical wiring diagram of the system is shown in Figure 17.

To assess the precision of the proposed AC outlet transient overvoltage peak model at the sending converter station during phase change failure and DC blocking faults in the multi-DC transmission system incorporating the turbine, phase change failure and DC blocking faults were intentionally induced at system 1 at 1 s. Subsequently, the voltage dynamics following the occurrence of the faults were thoroughly analyzed. The parameters of the verification system model are detailed in Table 11. The simulation analysis results and model calculation results are presented in Table 12, while Figure 18 illustrates the voltage dynamic process subsequent to the faults. This comprehensive verification procedure serves to validate the accuracy and reliability of the proposed method.

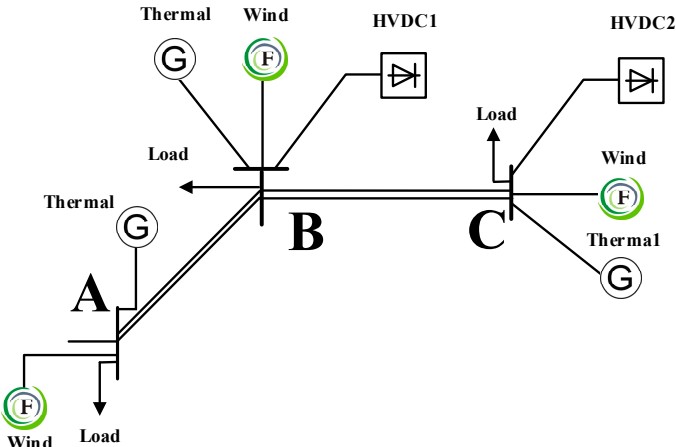

**Figure 17.** Geographical wiring diagram of the system.

**Table 11.** Equivalent parameters of the system.

| Parameters | Value |
|---|---|
| System 1 equivalent impedance $X_1$/p.u. | 0.0549 |
| System 2 equivalent impedance $X_2$/p.u. | 0.0531 |
| Equivalent contact impedance $X_{12}$/p.u. | 0.1911 |
| Number of fans in the near zone 1/unit | 510 |
| Number of fans in the near zone 2/unit | 340 |

**Table 12.** Verification of system fault overvoltage peaks.

| Fault Type | Peak Value of CF/p.u. | | Error/% | Peak Value of DCB/p.u. | | Error/% |
|---|---|---|---|---|---|---|
| | Simulation Results | Calculation Results | | Simulation Results | Calculation Results | |
| CF | 1.0937 | 1.0967 | 0.27 | 1.0710 | 1.0597 | 1.06 |
| DCB | 1.1013 | 1.1132 | 1.08 | 1.0619 | 1.0711 | 0.87 |

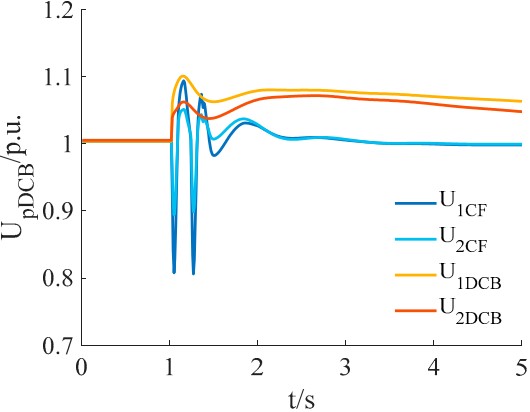

**Figure 18.** Verify the voltage dynamic curve after system failure.

From the analysis, it can be seen that the calculated results and the peak error of the transient simulation model results are small, and the accuracy of the mathematical model meets the requirements.

## 4. Conclusions

This paper introduces an analytical approach that aims to determine the transient peak value resulting from two types of faults: DC blocking (DCB) and commutation failure (CF). In a single-sending HVDC system, the transient overvoltage is primarily influenced by the short-circuit capacity and the surplus reactive power associated with DCB and CF faults. However, in a multi-sending HVDC system, it is crucial to consider the impact of DC-DC interaction on the transient voltage behavior.

When wind turbines are located in close proximity to the fault, the occurrence of DC blocking does not trigger a LVRT response in the turbines. This is because the converter station becomes the sole reactive source responsible for mitigating the transient overvoltage. On the other hand, commutation failure leads to multiple wind turbines entering LVRT. The LVRT characteristics of these wind turbines transform the wind farm into an additional reactive power source, acting in conjunction with the converter station. This collaboration between the two reactive power sources intensifies the transient overvoltage. In a single-send HVDC system, the overvoltage level is predominantly influenced by the short-circuit ratio. An increased short-circuit ratio leads to a reduction in the overvoltage level. Conversely, in a multi-send HVDC system, the overvoltage level is determined by the equivalent impedance of each individual system. In DC systems with turbines located in the vicinity of the DC zone, the overvoltage level at the converter bus is affected by the power characteristics of the turbines during LVRT.

To validate the proposed calculation method and accurately assess the severity of overvoltage, simulation results are provided. The results demonstrate the effectiveness of the proposed approach in accurately evaluating the extent of overvoltage and its implications.

The simplification employed in this paper for the AC system involves representing it as an electromotive force in series with an impedance. However, this analytical approach introduces certain inaccuracies. It is imperative to conduct additional research in the future to comprehensively examine the impact of various factors on the excitation characteristics of AC system generators. Furthermore, apart from investigating the influence of LVRT characteristics in wind turbines, this paper solely focuses on the effects of DC near-zone turbines. Therefore, it is crucial to extend the analysis to encompass the influence of wind farms operating as decentralized distribution systems on overvoltage.

**Author Contributions:** Conceptualization, F.L. and B.Q.; methodology, F.L. and B.Q; software, F.L. and B.Q.; validation, F.L. and B.Q.; formal analysis, F.L., B.Q. and X.H.; investigation, F.L. and B.Q.; resources, F.L. and B.Q.; data curation, F.L., B.Q., D.Z., X.S. and M.W.; writing—original draft preparation, F.L. and B.Q.; writing—review and editing, F.L., B.Q., D.L., Z.L., M.W. and H.Y. All authors have read and agreed to the published version of the manuscript.

**Funding:** This research received funding from the Science and Technology Project of the State Grid Corporation of China (a study of power grid transient stability support capability evaluation and enhancement method under high penetration of renewable energy, 5108-202218280A-2-280-XG).

**Institutional Review Board Statement:** Not applicable.

**Informed Consent Statement:** Not applicable.

**Data Availability Statement:** The data presented in this study are available in the article.

**Conflicts of Interest:** The authors declare no conflict of interest.

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
