# Peer review of "Calculation Method of DC Fault Overvoltage Peak Value for Multi-Send HVDC Systems with Wind Power"

_electronics, doi:10.3390/electronics12143157_

Round 1

Reviewer 1 Report

This article focuses on a method for analyzing the peak overvoltage at the converter bus resulting from DC faults in multi-send HVDC systems. This method considers the influence of DC/DC coupling and wind turbine low voltage ride through (LVRT) characteristics on overvoltage, which can calculate the peak overvoltage following a DC fault without the need for complex modeling or dynamic simulation software.

Although the paper is well organized, some problems need to be clarified as follows.

1.     The improved values or outstandingly data can be shown in the sections of ABSTRACT and CONCLUSIONS.

2.     Table 3 is used to show the overvoltage peak value of DC blocking under different system short-circuit capacity. Case 5 has the highest error. Why?

3.     The fluctuation is more serious at X1=0.6 in Figure 11.

4.     The peak value of DCB is more state than that of CF in Table 4 despite simulation or calculation result. Why?

5.  In line 425, this is a serious typo. Table 11. This is a table. Tables should be placed in the main text near to the first time they are cited. It needs to be revised.

6.     The simulation tools and software need to be demonstrated.

7.     The background of a single-sending HVDC system can be detailed in introduction.

8.     It can make a flow to describe the compared meaning for various X1, X2, and X12.

Reviewer 2 Report

Title of the peer-reviewed manuscript: “Calculation Method of DC Fault Overvoltage Peak Value for Multi-send HVDC Systems with Wind Power”.

The manuscript consists of an Abstract, Keywords, an Introduction section, three main parts, a Conclusion section, list of References from 26 titles, 15 of which were published during the last 5 years. The manuscript contains 17 Figures and X Tables.

The purpose of the study is to improve the stability of power systems by reducing the probability of failures. In the paper, the authors proposed a simplified method for analyzing the peak overvoltage on the bus resulting from DC faults  in multi-channel high-voltage direct current (HVDC) systems integrated with wind farms. The results of the presented study are of high practical importance for the design and operation of power systems.

The studies were carried out by mathematical modeling using the PSCAD and PSD-BPA software.

Questions and recommendations:

1. In my opinion section 2 Problem formulation should be included in section 1 Introduction.

2. Requires an explanation of how the results presented in Figures 3, 4 and 5 were obtained.

3. For the purpose of verification, the authors compare the calculation results with the simulation results. However, the results are usually confirmed by comparison with the results of a direct natural experiment. Have the authors conducted experimental studies? If not, then it requires an explanation on what the authors' confidence in the adequacy of the simulation results is based.

4. In the Conclusion section, the results obtained in the paper should be presented more clearly. I would also recommend that the authors provide plans for their future research.

In general, the manuscript is well-written, and the material presented in it is of interest to scientists, researchers and specialists in the field of electric power industry. I recommend this article for publication after minor revision.

Reviewer 3 Report

SUMMARY

The article introduces an analytical approach to determine the transient peak value resulting from two types of faults in high voltage direct current (HVDC) systems: DC blocking (DCB) and commutation failure (CF). The study considers the impact of DC-DC interaction and the collaboration between the converter station and wind turbines in multi-sending HVDC systems. The article proposes a calculation method to accurately assess the severity of overvoltage, and simulation results are provided to validate the approach. The results demonstrate the effectiveness of the proposed method in accurately evaluating the extent of overvoltage and its implications.

POSITIVE ASPECTS

1. Based on a literature review, the authors identified the gaps in knowledge and the need for a comprehensive overvoltage assessment methodology for multi-send HVDC systems incorporating wind turbines.
2. Based on a literature review, the authors mentioned the factors that influence transient overvoltage in multi-send HVDC systems, including the strength of the AC system, the electrical distance between systems, and DC power.
3. The authors develop an analytical expression for transient overvoltage in a multi-send HVDC system while analyzing the impact of overvoltage in such a system.
4. The authors derived algebraic equations to describe the power flow in the system and analyzed the voltage vector relationship between different buses after a fault.
5. The authors combined analytical expressions, mathematical equations, and theoretical analysis methods to investigate the transient overvoltage behavior in a multi-send HVDC system integrated with a wind farm, considering factors such as DC-DC coupling, LVRT capability, and wind turbine characteristics.
6. The authors conducted several analyses to evaluate the proposed method for calculating transient overvoltage in HVDC systems. The case studies demonstrated the effectiveness and accuracy of the proposed method in calculating overvoltage in HVDC systems, considering different scenarios and parameters.

CONCERNS

The presented work is useful but has some issues that need to be removed. Points that must be addressed by authors are listed below:

Major concerns
1. Authors should write out the meaning of the acronyms GE, T1, CF, and DCB upon the first usage.
2. According to ISO 80000-1: 2009 standard, the symbols for units are always written in Roman (upright) type, irrespective of the type used in the rest of the text. The unit symbol shall remain unaltered in the plural and is not followed by a full stop except for normal punctuation, e.g., at the end of a sentence.
3. The text lacks a thorough explanation of the abbreviations UdrN, Rd, and ULN. In an academic context, it is important to provide a clear context and understandable definition of these abbreviations. In this case, the explanation of the abbreviations UdrN, Rd, and ULN need to be incorporated into the text to provide accurate information and ensure clear communication with readers.
4. In Tables 4 to 10 and Table 12, it is unclear what the authors indicate as the values of “Error.” In an academic context, it is crucial to provide precise and unambiguous explanations of the variables and terms used in tables and figures.
5. In line 422, the authors incorrectly refer to Figure 16. Please correct the erroneous reference to the figure.

Minor concerns
1. According to ISO 80000-1: 2009 standard, the symbol of the unit shall be placed after the numerical value in the expression for a quantity, leaving a space between the numerical value and the unit symbol (see, e.g. lines 310 to 318). Correct all text accordingly.

RECOMMENDATIONS

1. Please verify the description of Table 11 and make the necessary corrections accordingly.

QUESTIONS

I have one question for the authors of the article.
1. The authors claim in lines 429 to 431 that the calculated results and the maximum error of the transient simulation model results are small, and the accuracy of the mathematical model meets the requirements. What specific requirements for the accuracy of the mathematical model do the authors have in mind?

Answer the given questions with comments in the manuscript.

CONCLUSION

I find this article helpful. Regretfully, the paper cannot be accepted in its present form. The authors of the present article have to correct the issues.

Reviewer 4 Report

The paper “Calculation Method of DC Fault Overvoltage Peak Value for 2 Multi-send HVDC Systems with Wind Power” an analytical approach that aims to determine the transient peak value resulting from two types of faults: DC blocking and commutation failure is presented.

To analyze this calculation method and accurately assess the severity of overvoltage, simulation results are provided.  It is mentioned that the simulation results demonstrate the effectiveness of the proposed approach in accurately evaluating the extent of overvoltage and its implications.

The paper has the following structure: 

1. Introduction

2. Problem formulations

3. Analysis Method of Transient overvoltage

3.1. Theoretical analysis method for DC fault overvoltage

3.2. Overvoltage peak analysis method caused by DC faults

3.3. Analysis method on transient overvoltage cosidering LVRT of Wind farm

4. Case study

4.1. Case Study of transient overvoltage in single-send HVDC system

4.2. Analysis of the effect of the low voltage ride-through characteristics of the wind turbine on 336 the overvoltage peak

4.3. Case study of transient overvoltage in two-sending HVDC system

4.4. Case study of transient overvoltage considering LVRT of wind farm

5. Conclusions

Remarks

1. The paper considers an analytical method for 2 systems. Expansion to more systems is not discussed.

2. Revision of the text for more clarity is required. Remarks are marked in yellow and red in the attached file.

3. To attract more new readers, you can include a drawing with a map of the region and diagrams of grid systems.

4. It is necessary to expand the list of references to 35-40 sources.

 Moderate editing of English language required

Reviewer 5 Report

Please, see attached document

Round 2

Reviewer 1 Report

none

Author Response

Since the ‘Comments and Suggestions for Authors
‘ is ‘none‘, the authors has nothing to reply. 

Reviewer 3 Report

SUMMARY

The article introduces an analytical approach to determine the transient peak value resulting from two types of faults in high voltage direct current (HVDC) systems: DC blocking (DCB) and commutation failure (CF). The study considers the impact of DC-DC interaction and the collaboration between the converter station and wind turbines in multi-sending HVDC systems. The article proposes a calculation method to accurately assess the severity of overvoltage, and simulation results are provided to validate the approach. The results demonstrate the effectiveness of the proposed method in accurately evaluating the extent of overvoltage and its implications.

POSITIVE ASPECTS

1. Based on a literature review, the authors identified the gaps in knowledge and the need for a comprehensive overvoltage assessment methodology for multi-send HVDC systems incorporating wind turbines.
2. Based on a literature review, the authors mentioned the factors that influence transient overvoltage in multi-send HVDC systems, including the strength of the AC system, the electrical distance between systems, and DC power.
3. The authors develop an analytical expression for transient overvoltage in a multi-send HVDC system while analyzing the impact of overvoltage in such a system.
4. The authors derived algebraic equations to describe the power flow in the system and analyzed the voltage vector relationship between different buses after a fault.
5. The authors combined analytical expressions, mathematical equations, and theoretical analysis methods to investigate the transient overvoltage behavior in a multi-send HVDC system integrated with a wind farm, considering factors such as DC-DC coupling, LVRT capability, and wind turbine characteristics.
6. The authors conducted several analyses to evaluate the proposed method for calculating transient overvoltage in HVDC systems. The case studies demonstrated the effectiveness and accuracy of the proposed method in calculating overvoltage in HVDC systems, considering different scenarios and parameters.

CONCERNS

My comments are merely editorial (of minor type).

Minor concerns
1. According to ISO 80000-1: 2009 standard, the symbol of the unit shall be placed after the numerical value in the expression for a quantity, leaving a space between the numerical value and the unit symbol (see, e.g. lines 332, 342, and 343). Correct all text accordingly.
2. In line 214, you must separate the word “bus” from the abbreviation “P'si” with a space.

CONCLUSION

I find this article helpful. Regretfully, the paper cannot be accepted in its present form. The authors of the present article have to correct the issues.

Reviewer 5 Report

Please, find the comments in the attached file

Round 3

Reviewer 5 Report

A last comment regarding the "strength" parameter: The explanation given by the authors is satisfactory. However, that would be desirable to clarify in the paper to be published since readers may find the use of "strength" in the work ambiguous.
